# Hydrocarbon Resin-Based Composites with Low Thermal Expansion Coefficient and Dielectric Loss for High-Frequency Copper Clad Laminates

**DOI:** 10.3390/polym14112200

**Published:** 2022-05-28

**Authors:** Jiaojiao Dong, Hao Wang, Qilong Zhang, Hui Yang, Jianlin Cheng, Zhaoyue Xia

**Affiliations:** State Key Lab Silicon Mat, School of Materials Science and Engineering, Zhejiang University, Hangzhou 310027, China; 22026091@zju.edu.cn (J.D.); ustbwanghao@163.com (H.W.); yanghui@zju.edu.cn (H.Y.); chjianlin@outlook.com (J.C.); 22126039@zju.edu.cn (Z.X.)

**Keywords:** polymer-based composites, substrate materials, dielectric properties, fillers

## Abstract

The rapid development of the 5G communication technology requires the improvement of the thermal stability and dielectric performance of high-frequency copper clad laminates (CCL). A cyclic olefin copolymer (COC) resin was added to the original 1,2-polybutadienes (PB)/styrene ethylene butylene styrene (SEBS) binary resin system to construct a PB/SEBS/COC ternary polyolefin system with optimized dielectric properties, mechanical properties, and water absorption. Glass fiber cloths (GFCs) and SiO_2_ were used to fill the resin matrix so to reduce the thermal expansion coefficient (CTE) and enhance the mechanical strength of the composites. It was found that the CTE of polyolefin/GFCs/SiO_2_ composite laminates decreased with the increase of SiO_2_ loading at first, which was attributed to the strong interfacial interaction restricting the segmental motion of polymer chains between filler and matrix. It was obvious that the addition of COC and SiO_2_ had an effect on the porosity, as shown in the SEM graph, which influenced the dielectric loss (D_f_) of the composites directly. When the weight of SiO_2_ accounted for 40% of the total mass of the composites, the laminates exhibited the best comprehensive performance. Their CTE and D_f_ were reduced by 63.3% and 22.0%, respectively, and their bending strength increased by 2136.1% compared with that of the substrates without COC and SiO_2_. These substrates have a great application prospect in the field of hydrocarbon resin-based CCL.

## 1. Introduction

Nowadays, the industry of autopilot unmanned aircraft, the Internet of things (IOT) technology and the augmented reality/virtual reality (AR/VR) technology are emerging rapidly, promoting the gradual development of the electronic components of consumer electronics, computers, communication, and electronic equipment to achieve high speed, miniaturization, high integration [1]. To achieve ultra-reliable and low-delay communication, higher requirements are put forward for all electronic device carriers and component interconnection materials that carry signal transmission, conversion and recording functions. As an important part of these electronic devices, the printed circuit board (PCB) that provides electrical connection between electronic components needs not only higher integration but also greater data transmission capacity [2,3,4]. The material, stack-up design, channel design, power noise filtering, termination schemes of PCB will greatly affect the data transmission capacity [5,6,7]. It is well known that substrates with low dielectric constant (D_k_) and D_f_ can achieve characteristics of high fidelity and low delay in the process of high-frequency signal transmission.

Polyolefin resins are often used as high-frequency dielectric materials because of their characteristics of low polarizability and good dielectric performance [8,9,10,11]. The low polarity of C-H (the electronegativity of C is 2.5, that of H is 2.1) in the molecular chain of polyolefin resin and the conformation of the molecular chain arranged in a zigzag plane provide the resin with excellent dielectric properties [12,13,14]. Among these resins, PB is the ideal material for high-frequency CCL due to its low D_k_, low D_f_ and low toxicity. For instance, the RO 4000^®^ series based on PB polymer matrix from Rogers corporation exhibit good performance as high-frequency circuit boards [15]. Based on ternary composite materials (PB/styrene/butadiene/styrene triblock copolymer (SBS)/ethylene/propylene/dicyclopentadiene (EPDM)), Bo and Wu et al. found that appropriate crosslinking agents can more effectively enhance the degree of the crosslinking reaction in the material, and the modification of SiO_2_ microspheres can improve the interfacial compatibility of different phases [15,16,17,18]. Xuan and Zhang et al. designed a sandwich-structure composite material [19,20], with an intermediate layer of polytetrafluoroethylene (PTFE) and outer layers consisting of PB films or malefic–anhydride polybutadiene–amino-terminated polyamide oligomers films, achieving an ultra-low D_f_ (0.0012). All these developed copper-clad laminate materials with excellent performance have made important contributions to the field of high-frequency substrates. Nevertheless, the disadvantages of PB-based composites in terms of CTE and D_f_ limit their wide application as high-frequency substrates. There is an urgent demand for PB-based composite materials with excellent thermal and dimensional stability as well as dielectric properties. The mismatch of CTE between composites and copper foil deserves more attention. Otherwise, the copper foil will fall off from the surface of the CCL during application due to the mismatch between the size change of the two parts. This reduces the reliability and service life of the material and in some cases leads to catastrophic device failure. The high CTE of the substrates often impairs the through-hole stability of the CCL, the violent contraction and expansion usually ruptures the through-hole and thus breaks down the circuit during hot processing. Conversely, materials with CTE close to that of copper foil have excellent dimensional stability. In previous works, three main ways to reduce the CTE of substrate materials were proposed: firstly, the addition of an inorganic filler with low (such as SiO_2,_ carbon nanotubes) [21,22,23,24,25,26,27,28,29,30] or negative (such as Hafnium pyrovanadate) CTE [28,29] into the organic matrix; secondly, the addition of a low-CTE organic monomer (such as an alkoxysilyl-functionalized resin) or a negative-CTE organic monomer (such as dibenzocyclooctane) or the use of a resin with a low CTE (such as cyanate ester resin) [30,31,32,33]; finally, the interpenetration of the network structure by combining a ceramic skeleton with organic matter (such as Al_2_W_3_O_12_) [34,35]. However, the CTE is not the only factor that should be considered, and other key indicators such as D_k_ and D_f_ should be taken into account either. Considering the various performance of CCL materials, the high polarity of organic monomers or organic matter precludes their use in substrate materials requiring a low D_f_. In addition, designing a kind of inorganic–organic multiphase composite is the most common and easiest method to reduce the CTE. As a cheap and readily available filler, SiO_2_ is often used to complement polymers to enhance the overall thermal stability, dimensional stability and mechanical properties of the composites. Spherical SiO_2_ has outstanding disadvantages and could make composites more compact. Olefin is a typical low-polarity copolymer, so it can maintain low D_k_ and D_f_ at high frequency. Among olefins, cyclic olefin copolymer (COC) has an ultra-low D_k_ of 2.21 and an ultra-low D_f_ of 0.00021 at 10 GHz. Therefore, blending COC with PB would be an effective method to reduce the D_f_ of the substrates [36,37,38,39].

Based on previous studies, this paper adopted PB and SEBS as a resin matrix, introducing spherical SiO_2_ and COC to reduce the CTE and D_f_ and augmenting GFCs to optimize the mechanical performance of the material. The influence of COC and SiO_2_ content on the CTE and D_f_ of the composite was thoroughly studied, and the mechanical performance, thermal property and water absorption of the materials was also examined.

## 2. Experimental Procedures

### 2.1. Materials

PB (Mn:3200, vinyl content:92%), SEBS and COC were provided by Nippon Soda Co., Ltd. (Tokyo, Japan), Kraton Co., Ltd. (Houston, TX, USA) and TOPAS Advanced Polymers Co., Ltd. (Tokyo, Japan), respectively. The compounds 2,5-dimethyl-2,5-di (tert-butyl peroxyl) hexane (DBPH), cyclohexane and γ-methacryloxy propyl trimethoxy silane (KH570) were purchased from Aladdin Biochemical Technology Co., Ltd. (Shanghai, China). SiO_2_ particles were provided by Sanshiji New Material Technology Co., Ltd. (Hangzhou, China).

### 2.2. GFCs Surface Modification

The GFCs were modified by KH570 which can improve the surface adhesion of GFC and resin. Firstly, 10.5 g of water was added to 250 mL of alcohol to prepare a 95 wt% alcohol aqueous solution. A certain weight of KH570, which accounted for a 2% weight fraction of the GFC, was dispersed in the alcohol aqueous solution for the pre-hydrolysis of KH570. Then, the GFCs were immersed in the alcohol aqueous solution with KH570, followed by ultrasonic dispersion for 20 min. Finally, the modified GFCs were obtained by drying at 80 °C for 12 h.

### 2.3. Preparation of SiO_2_/PB/SEBS/COC Solutions

It has been proved that the amount of crosslinking agent has a great influence on the performance of CCL [18]. As shown in Appendix A, over 1.07 phr (0.3 g) DBPH has a negative effect on the dielectric properties. In this work, 3 g of SEBS, 10 g of PB, and 0.3 g of DBPH were placed into 75 mL of cyclohexane. In the same way, 100 mL of cyclohexane was used to dissolve from 0 g to 20 g of COC, in quantities different by 5 g (named C0, C5, C10, C15, C20), respectively. Then, the mixtures mentioned above were stirred rapidly at room temperature until the resin was totally dissolved. To obtain homogeneous PB/SEBS/COC solutions, the solutions were mixed and stirred for 2 h. Furthermore, SiO_2_ particles were added into the polyolefin solution at weight fractions of 0%, 10%, 20%, 30%, 40%, 50% (named S0, S1, S2, S3, S4, S5). The content of PB, SEBS, COC was 10 g, 3 g and 15 g, respectively. To prevent the settling of the SiO_2_ particles, the solutions were subjected to ultrasound for 20 min every 4 h and stirred magnetically for the remaining 12 h. Hereto, PB/SEBS/COC solutions with different amounts of COC and PB/SEBS/COC/SiO_2_ solutions with different amounts of SiO_2_ were prepared.

### 2.4. Synthesis of GFC-Reinforced Resin Composites

Modified GFCs (10 cm ×10 cm) were infused in the above prepared solutions until they were coated by the solution; then, they were dried at 80 °C for 15 min. After that, 10 layers of GFCs prepregs were placed between two layers of copper foil and placed into a thermocompression machine. The prepregs were pressed at a pressure of 3 MPa at 195 °C for 2 h. Finally, the hot-pressed laminates were cooled to room temperature. The process of preparation mentioned above is shown in Figure 1.

## 3. Characterization

The hydrophobicity of the modified GFCs was measured by a video-based contact angle measuring device (OCA 20, Data physics Ltd., Santa Clara, CA, USA). A laser particle analyzer (LS13320, Beckman Coulter Ltd., Brea, CA, USA) was used to reveal the particle size distribution of SiO_2_ powders. The morphology of the PB/SEBS/COC composites was investigated by field-emission scanning electron microscopy (FESEM, SU8010, Hitachi Ltd., Tokyo, Japan). The infrared spectra of GFCs were determined by Fourier-transform infrared spectrometry (FTIR, Nicolet 5700, Thermo Nicolet Ltd., Waltham, MA, USA) using the ATR mode. A differential scanning calorimetric device (DSC, Q200, TA Ltd., Prospect Park, PA, USA) was used to assess the glass transition temperature (T_g_) of the samples in a nitrogen atmosphere at 10 °C/min. Thermogravimetric analysis (TGA) was carried out using an instrument (TGA, Q500, TA Ltd., Prospect Park, PA, USA) to measure the decomposition temperature of the composites at a heating rate of 20 °C/min in a nitrogen atmosphere. The separated dielectric resonator method (SPDR) using an Agilent PNA-L network analyzer (N5234A, Keysight Technologies Ltd., Santa Rosa, CA, USA) was used to measure the D_K_ and D_f_ (detailed definitions are shown in supplementary material, and the shown values are the average value of three samples of the same type) of the samples in the microwave frequency region at room temperature. The moisture absorption of the samples was calculated after immersion in water at room temperature for 24 h. The CTE was obtained using a thermomechanical analyzer (TMA, Q400EM, TA Ltd., Prospect Park, PA, USA) by recording the dimension change with temperature. A universal material testing machine (CMT5205, MTS Ltd., Eden Prairie, MN, USA) was employed to test the bending strength by the three-point bending resistance method. The thermal conductivity was determined by Hot Disk TPS 2500 S (Uppsala, Sweden) in the isotropic pattern. The thickness of different samples was tested and recorded as shown in Appendix A.

## 4. Result and Discussion

### 4.1. Microstructure Analysis

As shown in Figure 2a, SiO_2_ exhibited regular sphericity and uniform size distribution. Figure 2c shows the mean diameter of the SiO_2_ microspheres was about 1.3 µm, which is consistent with the value estimated in Figure 2a. Two peak distributions of the SiO_2_ spheres’ size within the limits of 0.5–8 µm were also observed. The cross-sectional morphology of the composite with the SiO_2_ filler and the resin matrix is shown in Figure 2b, which indicates that the filler was uniformly distributed in matrix.

Figure 1 shows that the GFCs were modified by KH570, whose chemical structure is shown in Figure 2d. Figure 2f shows the FTIR spectra of untreated and modified GFCs. The process of modification occurred in three stages. Firstly, -Si(OCH_3_)_3_ was hydrolyzed into -Si(OH)_3_. Secondly, some -SiOH bonds reacted with hydroxyl radicals on the surface of GFCs, then -SiO-G (G indicates the surface of GFCs) formed after dehydration and condensation. The characteristic absorption peaks of the hydroxyl (about 3300 cm^−1^) of the modified GFCs illustrated in Figure 2f was stronger than that of the untreated GFCs, indicating that KH570 had reacted with the hydroxyl groups on the surface of GFCs, which led to the reduction of the hydroxyl groups. Lastly, the other -SiOH bonds associated with each other and formed a web-like film which covered the GFCs [40]. The adhesive property of GFCs and resin were improved by the organic adsorption layer formed on the surface of the GFCs. Figure 2f shows a strong absorption peak at 1060 cm^−1^ representing the stretching vibration absorption of the -Si-O bond, whereas the symmetric stretching vibration absorption of the -Si-O-Si bonds is represented by the peaks at 800 cm^−1^ and 465 cm^−1^. In addition, the spectra of modified GFCs exhibited the carbonyl symmetric vibration absorption peak at 1716 cm^−1^. The contact angle between water droplets and the surface of GFCs is about 86° in Figure 2d and 123° in Figure 2e, which means that the nature of the GFCs changed from hydrophilic to hydrophobic after their modification. TG curves of untreated and modified GFCs are shown in Figure 2g. The weight loss below 200 °C was mainly caused by adsorbed water. The mass loss rate of untreated GFCs was higher than the loss of modified GFCs. Because the surface hydrophobicity of the modified GFCs increased, the adsorbed water decreased on the surface. Beyond 200 °C, the weight loss rate of GFCs modified by KH570 increased significantly, and this weight loss (about 2 wt%) corresponded to the combustion of KH570 on GFCs surface, indicating that about 2 wt% KH570 was coated on GFCs surface.

The cross sectional and surface SEM images of the hydrocarbon resin-based composites with different contents of COC and filled with different amounts of SiO_2_ are presented in Figure 3. As shown in Figure 3a, the amount of polyolefin could not cover the GFCs completely. Therefore, some voids appeared on the surface of C0, which led to an uncompacted structure. In contrast, when the amount of COC reached 15 g, as shown in Figure 3b, the GFCs were completely invisible, and the surface was smooth and compact. It is very clear in Figure 3f that there was excessive COC around the GFCs compared to Figure 3d,e, leading to an uneven surface, as depicted in Figure 3c. To strengthen the adhesive properties of polymer and GFCs and enhance the comprehensive performance of the materials, it is very important to avoid pores as much as possible. Contrary to the voids appearing in Figure 3d, the glass fibers were glued to each other closely by the resin, as shown in Figure 3e. An appropriate amount of SiO_2_ is one of the major reasons for the low porosity, because the spherical particles of SiO_2_ fill the gaps between the glass fibers in the GFCs, forming a dense structure. However, excessive SiO_2_ would result in a barrier effect, making the resin disperse, as shown in Figure 3i. A dispersed resin negatively affects the property of composites (compare with Figure 3h).

### 4.2. Dielectric Properties

Figure 4a,b show that the D_k_ of the complex substrate material was about 3.5. With the content of COC increasing, the D_f_ of the composites decreased continuously, as depicted in Figure 4a, dropping from 0.0028 to less than 0.0021, which was attributed to the relatively low D_f_ of COC (0.00021) and the denser structure. As depicted in Figure 4b, the D_f_ decreased at first and then increased with the increase of SiO_2_ content. Porosity, second phase, impurity and defects increase the non-intrinsic loss of dielectric materials. Figure 3g, h show that the voids in the composites were reduced as the content of SiO_2_ increased. Hence, the D_f_ of the composites decreased as the pores decreased. Many previous works showed that porosity is closely related to the D_f_ of composites [41]. However, adding excessive SiO_2_ destroyed the continuous structure of the resin and led to an increase of the voids in the composites, as shown in Figure 3i, thus increasing the D_f_. In conclusion, an appropriate content of SiO_2_ is crucial for the decrease of D_f_. The minimum D_f_ (0.0019) was obtained for the sample with 40 wt% SiO_2_ (S4) and was lower than that of current commercial CCL.

### 4.3. Thermal Stability

As shown in Figure 5a, the CTE of the substrate material increased with the increase of COC content, which was due to the high CTE of COC itself (about 60 ppm/°C). As shown in Figure 5b, when the appropriate amount of SiO_2_ with CTE of around 4 ppm/°C was added, the CTE of the materials was reduced steadily as filler loading proceeded, and once the content of SiO_2_ exceeded about 40 wt%, the CTE began to rise. This variation can also be obtained from a theoretical calculation formula of CTE [42]:(1)$$\alpha_{c}{=a+k}_{0}\varphi_{f}\varphi_{m}\left[\varphi_{f}\alpha_{f}{+\varphi }_{m}\alpha_{m}{+K}_{0}\varphi_{f}\varphi_{m}\left(\alpha_{f}+\alpha m\right){+k}_{0}\varphi_{f}\varphi_{m}k_{1}\right]$$

Here, a, k_0_ and k_1_ are constants; α_c_, α_f_, and α_m_ are the CTE values of the composite, filler, and matrix, respectively; φ_f_, φ_m_ are the volume fraction of filler and matrix. There were strong interfacial interactions between the polymer matrix and the SiO_2_ filler, which could restrict the segmental motion of polymer chains. It is obvious that the interfacial phase volume is an important factor affecting the performance of the filler. From Equation (1), it can be concluded that the CTE of the composite is related to the CTE of the components, and the volume of the interfacial phase is tightly related to the content of filler. The volume of the interfacial phase became lager as the filler increased. When the filler content was low, the CTE of SiO_2_ was the dominant factor influencing the CTE according to Equation (1). When the filler content was high, the interfacial phase volume was the dominant factor influencing the CTE. That is why we can observe a downward trend and an upward trend in Figure 5b. The reason mentioned above in relation to Figure 3i indicated that too much SiO_2_ negatively affects the continuity of the resin, which also results in the decrease of the CTE. Figure 5c shows the same pattern as that in Figure 5b, i.e., the CTE decreased and then increased when the amount of SiO_2_ increased.

The determination of the glass transition temperature (T_g_) is very important because it indicates the critical service temperature of the laminates. As shown in Figure 6a,b changes in the ratio of COC to SiO_2_ had no obvious influence on T_g_, which was about 175 °C. Figure 6c,d show that the composites began to decompose in the temperature range of 400 °C to 450 °C. The decomposition temperatures of all composites were similar, with no significant difference, and were around 425 °C. SiO_2_ and GFCs own relatively higher thermal conductivity than the polymer matrix; therefore, SiO_2_ and GFCs can absorb heat easily and form thermal barriers surrounding a polymer, causing thermal decomposition of the polymer matrix at a higher temperature. More importantly, the most molecules of the resin changed their structure from linear to reticular after crosslinking under the action of high temperature and a cross-linking agent. All the above factors increased the decomposition temperature of the composites to more than 400 °C. The thermal decomposition in inert atmosphere followed only one process for all samples. When heating in inert atmosphere, the final mass fractions of C0, C10, C20, S1, S3, S5 were 85%, 75%, 68%, 67%, 76%, 88%, respectively. Because the sample contained a hydrocarbon resin, glass fibers and SiO_2_ particles, the residue in the laminate which decomposed in an inert environment was a mixture of carbon, glass and SiO_2_. C0, C10, C20 produced the same residue (except SiO_2_) as S1. The decomposition temperature (T_d_, temperature at 5% mass loss) of the composites without SiO_2_ was about 465 °C, as shown in Figure 6c. The T_d_ of the composites with SiO_2_ changed from 463 °C to 470 °C, as shown in Figure 6d, which was due to the different content of SiO_2_, which has strong thermal stability.

Figure 6e, f show that the thermal conductivity (TC) of the materials was about 0.8 W/(m·K). Because the TC of COC and SiO_2_ particles is around 0.4 W/(m·K) and 1 W/(m·K), respectively, filling with SiO_2_ can improve the TC of the substrates. The TC of the laminates in this work will meet application requirements in the field of hydrocarbon resin laminates and is slightly higher than that of the present commercial substrates. As we know, the TC of Ro4003C from Rogers corporation and of R-5575 from Panasonic industry is 0.62 W/m·K and 0.6 W/m·K, respectively.

### 4.4. Mechanical Properties and Water Absorption

As shown in Figure 7a, the bending strength of the complex substrate without COC was only 11.23 MPa. With the increase of the COC content from 0 to 15 g, the bending strength increased to 204.55 MPa, an increase of 1721.5% compared to the bending strength of the sample without COC. It is shown in the Figure 3d,e that the increasing content of COC contributed to the densification of the samples, which is the main reason of the intensification of the bending strength of the sample. However, when the COC further increased, the bending strength of the composite decreased to 198.90 MPa. Adding excessive COC made the redundant resin coated the surface of the substrate. In the bending strength test, the pressure exerted by the test machine on the sample brings about compression deformation above the surface of the sample, whereas tensile deformation below the surface of the sample is borne by the surface resin. The surface resin without mechanical reinforcement (GFCs) was subjected to the testing strain, showing a deterioration of its mechanical properties. As depicted in Figure 7b, the bending strength of samples filled with SiO_2_ was higher than that of composites without SiO_2_. When the SiO_2_ loading was 30 wt%, the bending strength of the composites reached 251.21 MPa. The addition of the filler at an appropriate content enhanced the mechanical properties of the material because the SiO_2_ particles were evenly dispersed in the organic matrix. SiO_2_ particles and resin interact with each other with large friction. SiO_2_ particles, like tiny nails, fix the resin, which can be easily deformed, and prevent its excessive deformation, resulting in the enhancement of the bending strength of the composites. However, the particles come in contact with each other and continue to gather as the content of SiO_2_ increases. The phenomenon of stress concentration appears, so the mechanical properties of the whole composites decline.

As shown in Figure 7c, the water absorption of the composites was reduced and then increased as the COC was added, achieving an ultra-low value of 0.026%. The initial decrease of water absorption from 0.11% to 0.064% can be explained by the increasing relative density of the sample. However, the surface of the composites with excessive COC became uneven, as shown in Figure 3c. Uneven surfaces can hold more water, leading to increased water absorption. An appropriate amount of SiO_2_ densified the composites, and therefore the water absorption decreased from 0.067% to 0.026%. However, an excessive amount of SiO_2_ particles leads to the formation of agglomerates in the composites, which destroy its continuous structure, resulting in uncompacted samples. The increase in water absorption can be explained by the fact that the composites were less dense inside, as shown in Figure 7d.

## 5. Conclusions

Hydrocarbon resin-based composites comprising COC, PB, SEBS, SiO_2_ and GFCs were prepared. The introduction of an appropriate amount of COC will make the material denser, reduce D_f_ and water absorption, and improve the bending strength, while an excessive amount of COC will increase the water absorption and decrease the bending strength of the composites. The properties of the samples were also greatly affected by the SiO_2_ content. SiO_2_ entangles with macromolecular meshes, which hinders the movement of macromolecular segments, thus improving the thermal stability and mechanical properties of the samples. However, excessive SiO_2_ destroys the continuity of the resin, thus increasing CTE and D_f_. When the SiO_2_ content was 40 wt%, the substrate exhibited good dielectric properties (D_k_ of 3.28, D_f_ of 0.0018 at 10 GHz), good thermal stability (TC of 0.72 W/(m·K), CTE of 24.98 ppm/°C), good mechanical properties (bending strength of 237.35 MPa) and low water absorption (0.026%). The SiO_2_-filled PB/COC/SEBS composites have potential as high-frequency and high-speed substrates in various applications.

## Figures and Tables

**Figure 1 polymers-14-02200-f001:**
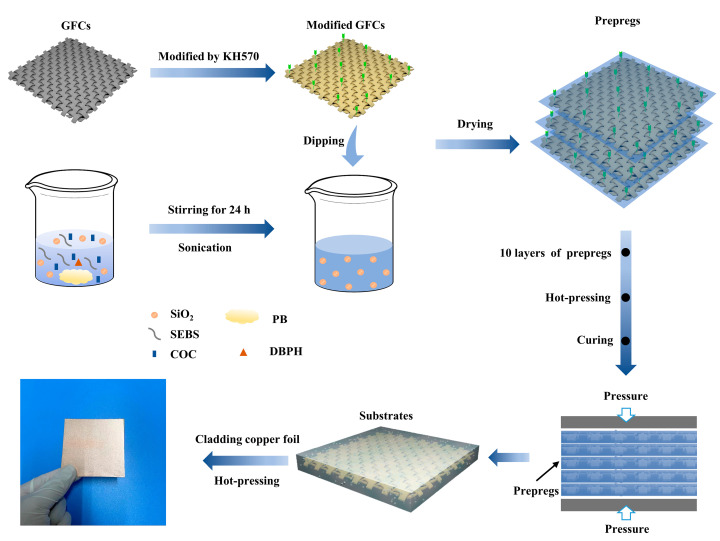
Process of sample preparation.

**Figure 2 polymers-14-02200-f002:**
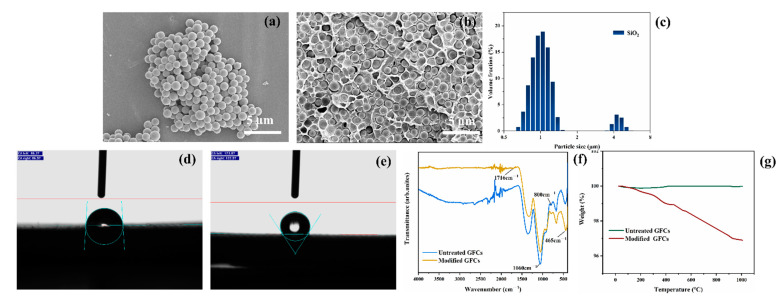
(**a**) SEM image of the spherical SiO_2_ powder; (**b**) cross-sectional micrograph showing neat polyolefin filling with 40 wt% SiO_2_; (**c**) particle distribution of SiO_2_; contact angle of water drop on (**d**) untreated GFCs and (**e**) modified GFCs; (**f**) FTIR spectra of untreated and modified GFCs; (**g**) TGA thermograms of untreated and modified GFCs.

**Figure 3 polymers-14-02200-f003:**
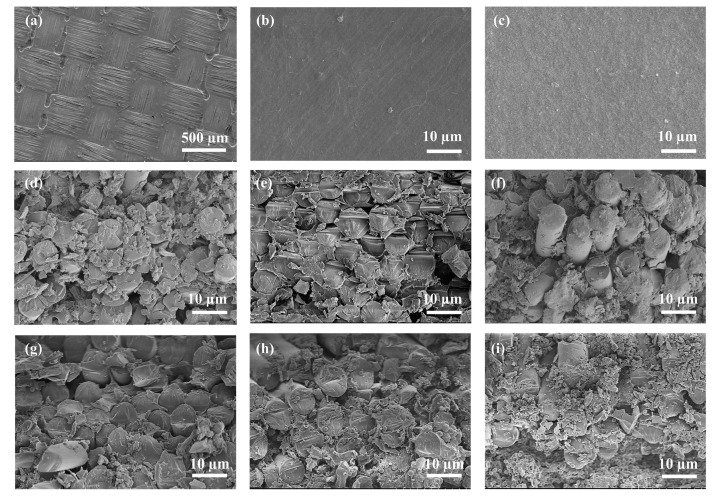
Surface morphology of different composites: (**a**) PB/SEBS/0 g COC (C0); (**b**) PB/SEBS/15 g COC (C15); (**c**) PB/SEBS/20 g COC (C20); cross-sectional morphology of different composites: (**d**) PB/SEBS/5 g COC (C5); (**e**) PB/SEBS/15 g COC (C15); (**f**) PB/SEBS/20 g COC (C20); (**g**) PB/SEBS/COC/10 wt% SiO_2_ (S1); (**h**)PB/SEBS/COC/30 wt% SiO_2_ (S3); (**i**) PB/SEBS/COC/10 wt% SiO_2_ (S5).

**Figure 4 polymers-14-02200-f004:**
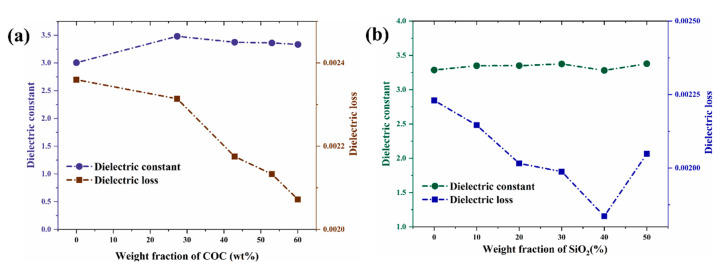
Influence of the content of COC (**a**) and SiO_2_ (**b**) on D_k_ and D_f_ at 10 GHz.

**Figure 5 polymers-14-02200-f005:**
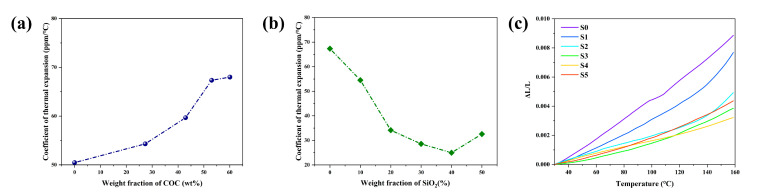
Influence of the content of COC (**a**) and SiO_2_ (**b**) on the CTE in the temperature range of 25–150 °C; (**c**) dynamic mechanical curves of S0, S1, S2, S3, S4, S5 as a function of the temperature.

**Figure 6 polymers-14-02200-f006:**
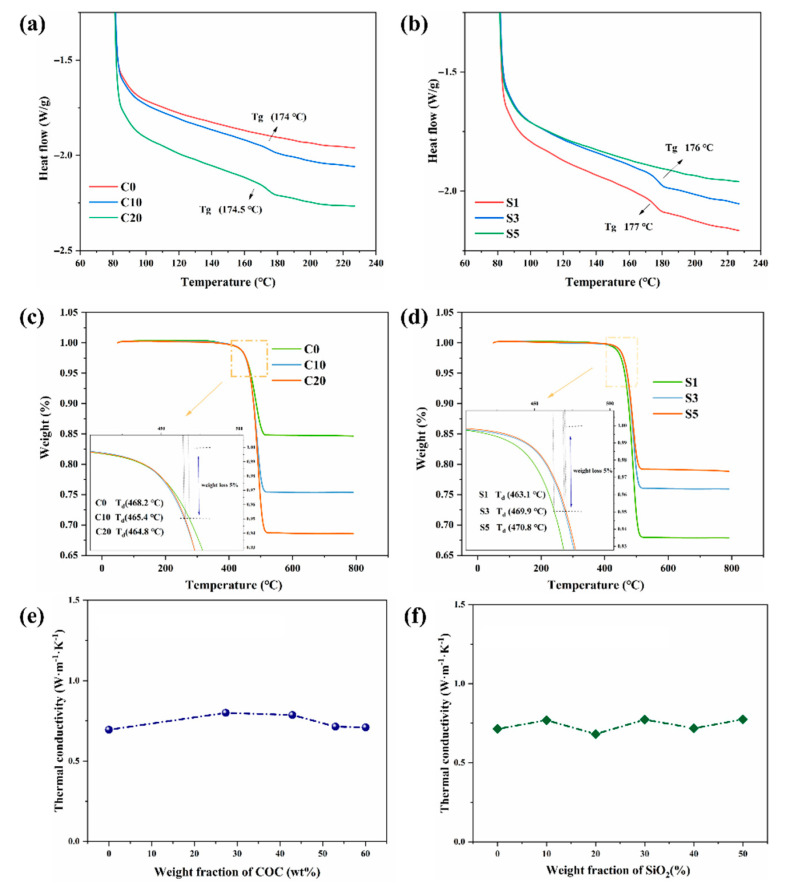
DSC curves of (**a**) C0, C10, C20 and (**b**) S1, S3, S5; TGA thermograms of (**c**) C0, C10, C20 and (**d**) S1, S3, S5; influence of the content of COC (**e**) and SiO_2_ (**f**) on the in-plane thermal conductivity.

**Figure 7 polymers-14-02200-f007:**
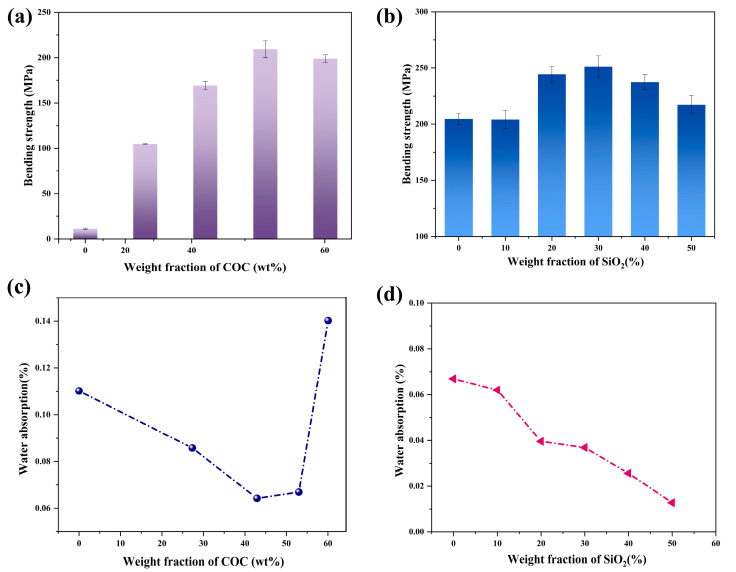
Influence of the content of COC (**a**) and SiO_2_ (**b**) on the bending strength; water absorption of the samples (**c**) with different content of COC and (**d**) with different content of SiO_2_.

## Data Availability

The data presented in this study are available from the corresponding author on reasonable request.

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
