# Peer review of "Hydrocarbon Resin-Based Composites with Low Thermal Expansion Coefficient and Dielectric Loss for High-Frequency Copper Clad Laminates"

_polymers, 2022, doi:10.3390/polym14112200_

Round 1

Reviewer 1 Report

Article Review

Title of the reviewed article: Hydrocarbon resin-based composite with low thermal expansion coefficient and dielectric loss for high frequency copper clad laminate

Overall, the manuscript is well written, the topic is interesting, and its solution is very beneficial for many experts in this field. Thus, I recommend the article for publishing with major modifications, which I subsequently mention.

  • Please define more precisely the parameters dielectric constant (Dk) and dielectric loss (Df) – are these the real and imaginary parts of the frequency and temperature dependent relative permittivity?
  • Almost all graphs are very difficult to read in the basic view - small size of the axis labels – please improve the readability of the figures even in the basic document view (A4 format).
  • I understand the inclusion of a higher number of citations in brackets [2-7] for the sentence on the tenth line of the first paragraph of the introduction, but I would have expected a slightly more detailed explanation of the term "data transmission capacity" – splitting the citations into several sub-brackets – just expand the text into two lines. I would have expected the same modification for the reference to multiple citations in brackets [8-14] in the second line of the second paragraph of the introduction and a more detailed specification of the "character of low in polarizability and good dielectric performance" in the text.
  • The introduction states "Ro 4000@" – the symbol ® should be used instead of the at-sign and the capital letter O instead of o.
  • The introduction mentions " (PB/SBS/ethylene-propylene-dicyclopentadiene (EPDM)) " – the abbreviation SBS is not explained in the text, only the abbreviation SEBS is explained – please explain. Please try to use only one bracket and do not insert additional sub-brackets.
  • Please edit the format of the text "et al[19,20]designed".
  • When providing the information "achieving ultra-low Df", I would ask you to add the numerical values that you consider ultra-low.
  • In the text "filler with low[21-30] or negative CTE[28, 28, 29]" you repeat twice in one bracket the citation 28. I would recommend giving specific examples of fillers with low and negative CTE values.
  • You define in the section Materials each material supply company a little differently – please follow the consistent format of "Kraton Co., Ltd. (Dover, OH, USA)".
  • You use the combination of number and unit inconsistently in the text (e.g., temperatures 80°C/80 °C) – please unify.
  • It is not specified which mode of FTIR technique was used – I assume ATR mode (for the spectra presented in Fig. 2g) – please add a specification.
  • For thermal analyses, I would not specify "at a scanning rate of 10 ℃/min", but only write about the rate of temperature change during the analysis (heating rate). For thermal analyzers (DSC, TGA, TMA) there is insufficient specification of the used instruments – manufacturer, type, specification of the company's location.
  • The definition of the standard 'IPC-TM-650' is insufficient – the general reader is probably not familiar with this standard – It could be cited as a source or specified in the text.
  • A figure should not be included above the first reference to it in the text. It is not appropriate to start a chapter with a figure – please adjust the positioning of Figure 2.
  • Figure 2 represents the results of the materials characterization and should not, in my opinion, include a representation of the chemical structure of the modifier KH570 – the chemical structure should be listed in the chapter Materials, possibly with other chemical structures (not required). Please explain why the spectrograms in the region from about 2000 to 2500 cm-1 are so unsteady – Is this due to the crystal background within the ATR method?
  • Figure 3-7 is also referenced after the figures – please change the position of the figures.
  • The ranges of dielectric constant and dielectric loss (I use the loss factor) values in the graphs in Fig. 4 are relatively narrow and so the presented differences may just be because of the different samples measuring. What is the number of measured samples of each material within the dielectric properties analysis? If only one sample was measured for each modification (material type), I would emphasize this in the article and take it into account in the evaluation. It is not necessary to repeat the measurements with a higher number of samples, but the values should be better commented on in the context of possible inaccuracies. The article states that composites with ten layers of prepreg were prepared, but the resulting thickness of the samples is not mentioned – please add. The dielectric properties were measured for the prepared composites (complex substrates) also with copper foils? – If it so, a discussion needs to be made regarding the influence of metal layers on the measurements.
  • The heat flux trends from the DSC in Fig. 6 show a steep drop around 80°C, which characterizes the stabilization of the instrument at the beginning of the measurement – this drop should be removed from the graph – after that the y-axis scale can be significantly adjusted and, in my opinion, the shown cut-out is not necessary. This brings up the question, why is the DSC implemented only from 80°C?
  • In the case of the mass change graphs (Figs. 6c and 6d), the y-axis shows the mass change as a percentage – but the y-axis shows values from 0 to 1 instead of 0 to 100.
  • I would recommend, that the description of the results seen in Figs. 7c and 7d, may include some specific numerical value in the text.
  • There is a double space in "and water" (third line) at the end.
  • In the text you give the term "10 GHz" three times, once putting a space between the number and twice not – please unify.
  • Please clarify what is the Acknowledgements and what is the Funding so that your organization will declare the properly referenced project.

Author Response

Response to Reviewer 1 Comments

Point 1: Please define more precisely the parameters dielectric constant (Dk) and dielectric loss (Df) – are these the real and imaginary parts of the frequency and temperature dependent relative permittivity?

Response 1: Thanks for your suggestions. Under high frequency electric field, the displacement current produced by insulating material is not orthogonal to the direction of electric field, which consumes power and causes loss. Thus, at high frequencies, the relative permittivity of the material is complex number ε*:ε* = εʹ - jεʺ,The imaginary part εʺ represents the loss of the material at high frequency,tanδ is defined as the residual Angle δ tangent of the phase Angle between the applied sine wave voltage and the current passing through the substrate, tan δ = ε''/ε'. In engineering, this value is commonly used to characterize the loss of materials, known as the loss Angle tangent. In the experimental part of the paper, The complex number ε* are tested as Dk (dielectric constant) by SPDR (separated dielectric resonator) and Df (dissipation factor,also called dielectric loss) for Dissipation Angle tangent. The parameters dielectric constant (Dk) and dielectric loss (Df) was not define precisely in this paper which might comfuse readers,so this part has been explained in detail in the section of Supplementary Material.

Point 2: It is not specified which mode of FTIR technique was used – I assume ATR mode (for the spectra presented in Fig. 2g) – please add a specification.

Response 2: Thanks for your suggestions. Yes, ATR mode is used in the test and I have stated it in the paper according to your suggestion.

Point 3: Figure 2 represents the results of the materials characterization and should not, in my opinion, include a representation of the chemical structure of the modifier KH570 – the chemical structure should be listed in the chapter Materials, possibly with other chemical structures (not required). Please explain why the spectrograms in the region from about 2000 to 2500 cm-1 are so unsteady – Is this due to the crystal background within the ATR method?.

Response 3: Thanks for your suggestions. I agree that the chemical structure should not be listed in characterization and it had been removed from Fig. 2. It is well known that crystal background will seriously affect the flatness of infrared spectrum. After reviewing other works and repeated tests, this phenomenon is common. Thanks for your professional advice.

Figure 2. (a) The SEM image of spherical SiO2 powder; (b) Cross-sectional micrograph of neat polyolefin filling 40 wt% SiO2; (c) particle distribution of SiO2; (d) The chemical structure of KH570; Contact angle of water drop on the (e) untreated GFCs and (f) modified GFCs;(g) FTIR spectra of untreated and modified GFCs;(h) TGA thermograms of untreated and modified GFCs.

Point 4: Fig. 4 are relatively narrow and so the presented differences may just be because of the different samples measuring. What is the number of measured samples of each material within the dielectric properties analysis? If only one sample was measured for each modification (material type), I would emphasize this in the article and take it into account in the evaluation. It is not necessary to repeat the measurements with a higher number of samples, but the values should be better commented on in the context of possible inaccuracies. The article states that composites with ten layers of prepreg were prepared, but the resulting thickness of the samples is not mentioned – please add. The dielectric properties were measured for the prepared composites (complex substrates) also with copper foils? – If it so, a discussion needs to be made regarding the influence of metal layers on the measurements.

Response 4: Thanks for your suggestions. There are three measured samples of each material within the dielectric properties analysis and the average of them was presented in Fig. 4 and it had been stated in this paper. The thickness of the samples had been added in Supplementary Material. All the samples were tested without copper foil.

Table. S1. The thickness of different samples.

Samples

Thickness (mm)

C0

0.897

0.939

0.871

C5

0.841

0.822

0.853

C10

0.759

0.716

0.773

C15

0.773

0.794

0.751

C20

0.844

0.835

0.852

S0

0.847

0.843

0.860

S1

0.875

0.877

0.873

S2

0.925

0.935

0.915

S3

0.829

0.831

0.826

S4

0.918

0.912

0.924

S5

0.932

0.941

0.921

Point 5: The heat flux trends from the DSC in Fig. 6 show a steep drop around 80°C, which characterizes the stabilization of the instrument at the beginning of the measurement – this drop should be removed from the graph – after that the y-axis scale can be significantly adjusted and, in my opinion, the shown cut-out is not necessary. This brings up the question, why is the DSC implemented only from 80°C?

Response 5: Thanks for your suggestions. Fig. 6 had been modified according to your advice. After reviewing other works, it was found that the Tg of the substrate was higher than 100℃. In order to save the test time, the DSC implemented only from 80°C.

Figure 6. DSC curves of (a) C0, C10, C20 and (b) S1, S3, S5; TGA thermograms of (c) C0, C10, C20 and (d) S1, S3, S5; The content of COC (e) and SiO2 (f) dependence of the in-plane thermal con-ductivity.

Point 6:

I understand the inclusion of a higher number of citations in brackets [2-7] for the sentence on the tenth line of the first paragraph of the introduction, but I would have expected a slightly more detailed explanation of the term "data transmission capacity" – splitting the citations into several sub-brackets – just expand the text into two lines. I would have expected the same modification for the reference to multiple citations in brackets [8-14] in the second line of the second paragraph of the introduction and a more detailed specification of the "character of low in polarizability and good dielectric performance" in the text;

When providing the information "achieving ultra-low Df", I would ask you to add the numerical values that you consider ultra-low;

would recommend giving specific examples of fillers with low and negative CTE values;

Almost all graphs are very difficult to read in the basic view - small size of the axis labels – please improve the readability of the figures even in the basic document view (A4 format);

The introduction states "Ro 4000@" – the symbol ® should be used instead of the at-sign and the capital letter O instead of o;

Response 6: Thanks for your suggestions. All these mistakes and inappropriate expression had been modified according to your advices.

Point 7:

The introduction mentions " (PB/SBS/ethylene-propylene-dicyclopentadiene (EPDM)) " – the abbreviation SBS is not explained in the text, only the abbreviation SEBS is explained – please explain. Please try to use only one bracket and do not insert additional sub-brackets;

Please edit the format of the text "et al[19,20]designed";

In the text "filler with low[21-30] or negative CTE[28, 28, 29]" you repeat twice in one bracket the citation 28;

You define in the section Materials each material supply company a little differently – please follow the consistent format of "Kraton Co., Ltd. (Dover, OH, USA)";

You use the combination of number and unit inconsistently in the text (e.g., temperatures 80°C/80 °C) – please unify;

For thermal analyses, I would not specify "at a scanning rate of 10 ℃/min", but only write about the rate of temperature change during the analysis (heating rate). For thermal analyzers (DSC, TGA, TMA) there is insufficient specification of the used instruments – manufacturer, type, specification of the company's location;

Response 7: Thanks for your suggestions. All these mistakes and inappropriate expression had been modified according to your advices. But some information of material supply companies can not be found , so they had been modified by the format of Kraton Co., Ltd. (USA).

Point 8:

The definition of the standard 'IPC-TM-650' is insufficient – the general reader is probably not familiar with this standard – It could be cited as a source or specified in the text;

A figure should not be included above the first reference to it in the text. It is not appropriate to start a chapter with a figure – please adjust the positioning of Figure 2;

Figure 3-7 is also referenced after the figures – please change the position of the figures;

In the case of the mass change graphs (Figs. 6c and 6d), the y-axis shows the mass change as a percentage – but the y-axis shows values from 0 to 1 instead of 0 to 100;

I would recommend, that the description of the results seen in Figs. 7c and 7d, may include some specific numerical value in the text;

There is a double space in "and water" (third line) at the end; In the text you give the term "10 GHz" three times, once putting a space between the number and twice not – please unify;

Please clarify what is the Acknowledgements and what is the Funding so that your organization will declare the properly referenced project.

Response 8: Thanks for your suggestions. All these mistakes and inappropriate expression had been modified according to your advices.

Reviewer 2 Report

The work «Hydrocarbon resin-based composite with low thermal expansion coefficient and dielectric loss for high frequency copper clad laminate» is devoted very modern topic. The description of problems to be solved is given in detail in Introduction. In addition, the choice of research objects is well justified on the basis of previous studies. The experimental section is well structured and illustrated. All figures in the text of the article are well designed and informative, but they are too small for effective analysis of the data presented. Research well thought out and executed. Many interesting results were obtained during the study. But I have some comments:

  1. Figure 4 a and b need to be combined into one.
  2. Figure 6 e and f need to be combined into one.

Author Response

Response to Reviewer 2 Comments

Point 1: All figures in the text of the article are too small for effective analysis of the data presented. Research well thought out and executed.

Figure 4 a and b need to be combined into one.

Figure 6 e and f need to be combined into one..

Response 1: Thanks for your suggestions. The size of figures had been modified to be convenient for reading and analysis. However the meaning of x-axis of Fig. 4. (a) and Fig. 4. (b) (the content of COC and SiO2 )are too different to be combined into one, so as to Fig. 6.(e) and Fig. 6.(f). It may comfused readers if they are combined into one figure with two x-axis.

Round 2

Reviewer 1 Report

Thank you for implementing all edits in the article that match most of my comments in the first review. The level of the article is definitely higher, and the text is more clearly readable. Formally, you have appropriately unified the characterization of the instruments and citation of too much literature in one place. The readability of some graphs is still a little lower for me, but overall, the graphs are adjusted for better readability. I already recommend the paper for publication, and I believe that at least some of the information presented in the paper will be useful for other scientists and will in turn advance the research further.